

# Characterizing uncertainty in shear wave velocity profiles from the Italian seismic microzonation database

5  Federico Mori[1], Giuseppe Naso[2], Amerigo Mendicelli[1], Giancarlo Ciotoli[1], Chiara Varone[1], Massimiliano Moscatelli[1]

[1]C.N.R. – Istituto di Geologia Ambientale e Geoingegneria, Area della Ricerca di Roma 1- Montelibretti, Via Salaria km 29,300 - 00015 Monterotondo, RM, Italy
10  [2]Dipartimento della Protezione Civile (DPC), via Vitorchiano 2, 00189 Rome, Italy

*Correspondence to*: Federico Mori (federico.mori@cnr.it)

**Abstract.** This research uses a large dataset from the Italian Seismic Microzonation Database, containing nearly 15,000 measured shear wave velocity (Vs) profiles across Italy, to investigate the uncertainties in seismic risk assessment. This extensive collection allows a detailed study of the seismic properties of soil with unparalleled precision. Our focus is on 15  evaluating Vs variations with depth within uniformly clustered areas, known as seismic microzones. These zones are carefully identified based on their spatial correlation and homogeneity in geological, geophysical, and geotechnical characteristics, which are critical for accurate prediction of seismic response. We contrast these results with clusters formed purely based on geographic survey density (here defined geographic clusters), thereby assessing the depth of our understanding of the subsurface geological and geophysical context. These results were further compared with those reported in the seismic code 20  and literature. This study of depth-dependent Vs variations helps to refine our models of subsurface seismic behaviour. Our main discoveries show that: 1) uncertainties associated with seismic microzones (geological and geophysical clusters) are consistently lower than those identified in geographic clusters, particularly in the first 30 m of depth; 2) Vs profile variations show negligible increases in uncertainty within a certain range of correlation distances (up to about 4,500 m); 3) uncertainties for seismic microzones are lower than those previously reported in seismic codes and in the literature, indicating the 25  effectiveness and precision of our methodological approach. The results of this study significantly improve local seismic response analysis and highlight the critical role of depth and spatial correlation in understanding seismic hazard. The dataset is available at https://doi.org/10.5281/zenodo.10885590 (Mori et al., 2024).

## 1 Introduction

A seismic microzonation project has been active in Italy since 2009 for the most vulnerable Italian municipalities, about 4,000 30  out of approximately 8,000 (Moscatelli et al., 2020). This project is based on the ability to map homogeneous zones with respect to the expected amplification of seismic ground motions (so-called Seismic Microzones, hereafter SM). These zones



must be sufficiently detailed to account for local features that influence ground motion and to identify earthquake-induced ground instabilities (e.g., liquefaction, landslides, surface faulting, soil compaction).

All the data collected in the Italian seismic microzonation project, both geographical (e.g., shapes of the SM and location of the surveys) and alphanumeric (geological, geotechnical, and geophysical parameters) were standardised according to the Seismic Microzonation Working Group (2008) and stored in a database. The results of the seismic microzonation studies are freely available at the link: https://www.webms.it/servizi/stats.php (last access: March 2024). The database has proved its great potential over the years: it has been crucial for the development of the methodology used to produce the Vs30 (Mori et al., 2020), and the seismic amplification factor maps of Italy (Falcone et al., 2021). Gaudiosi et al. (2023) recently published a collection of shear modulus reduction and damping ratio curves primarily derived mainly from the database, while Varone et al. (2023) benefited from this dataset in assessing earthquake-induced liquefaction in Northern Italy. Shear wave velocity (Vs) profiles obtained from geophysical prospecting are a highly valuable component, along with other data. The collection contains over 23,500 Vs profiles, last updated in December 2022, evenly distributed throughout Italy.

Researchers have extensively investigated the variation of Vs with depth over time. Stewart et al. (2014) used the Savannah River dataset to examine the variations of Vs at various depths, building on previous work by Toro (1997). The suggested standard deviation for the natural logarithm of Vs ($\sigma lnVs$) is 0.15 for depths up to 50 m and 0.22 for depths beyond 50 m. In parallel, the US nuclear industry developed the SPID (Screening, Prioritization, and Implementation Details in EPRI-SPID, 1993) framework, which recommends $\sigma lnVs$ ranging of 0.25 for the first 15 m and 0.15 for greater depths, again depending on the amount of available data. Shi and Asimaki (2018) and Toro (1995, 2022) studied the detailed uncertainty in Vs profile datasets with the aim of proposing several randomisation models. Romagnoli et al. (2022) performed a statistical analysis on about 3,500 Vs profiles within the Italian territory to evaluate the variability of Vs values in the engineering geological units of the subsurface, as outlined in the seismic microzonation studies by Seismic Microzonation Working Group (2008).

The results of previous studies show an apparent complexity in the spatial structure of the Vs parameter and divergent trends in its uncertainty with depth, highlighting the need for further investigation in this area. Following these studies, the present work uses the considerable amount of information available from the Italian seismic microzonation database to determine the uncertainty associated with the variability of Vs with depth. In particular, the vertical profiles of $\sigma lnVs$ are calculated and presented.

As a by-product, this work also provides for the first time the largest database of Vs profiles (about 15,000 refined profiles out of approximately 23,500 available) ever distributed worldwide, derived exclusively from geophysical surveys (see Mori et al., 2024, https://doi.org/10.5281/zenodo.10885590). The flow chart in Fig. 1 highlights the steps followed in this study to calculate the $\sigma lnVs$ values of the Vs profiles from the Italian seismic microzonation database. The steps can be summarised as follows:

1. build a robust and reliable **dataset of Vs profiles** by removing any errors or duplicates (section 2.1);
2. define the **range values** in the **spatial correlation analysis** between the Vs profiles surveys (sections 2.2 and 3.1);
3. check the distribution pattern of the data (i.e., **clustering**) using Moran's Index (sections 2.3 and 3.1);


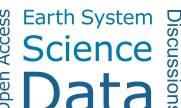

4.   define **geographic density-based clusters** (**GC**, HDBSCAN method, sections 2.3 and 3.1) and **geological-geophysical based clusters** (**SM**, seismic microzonation studies, sections 2.3 and 3.1);

5.   calculate the **σlnVs** of clusters formed by means of range value and GC (section 3.2);

6.   calculate the **σlnVs** of clusters defined by means of range value and SM (section 3.2);

7.   compare the results obtained from the two clustering methods and determine the approach that provides **lower σlnVs**
**values** (section 3.2);

8.   compare the obtained lower σlnVs values with those implemented in **seismic codes or reported in existing literature** (section 3.3).

The results demonstrate the effectiveness of the analysis performed on SM clusters: the standard deviation values of these clusters are significantly lower for all depths examined. These results are crucial for improving the randomisation of Vs
profiles in numerical simulation codes used for surface seismic response calculations and hazard analysis.

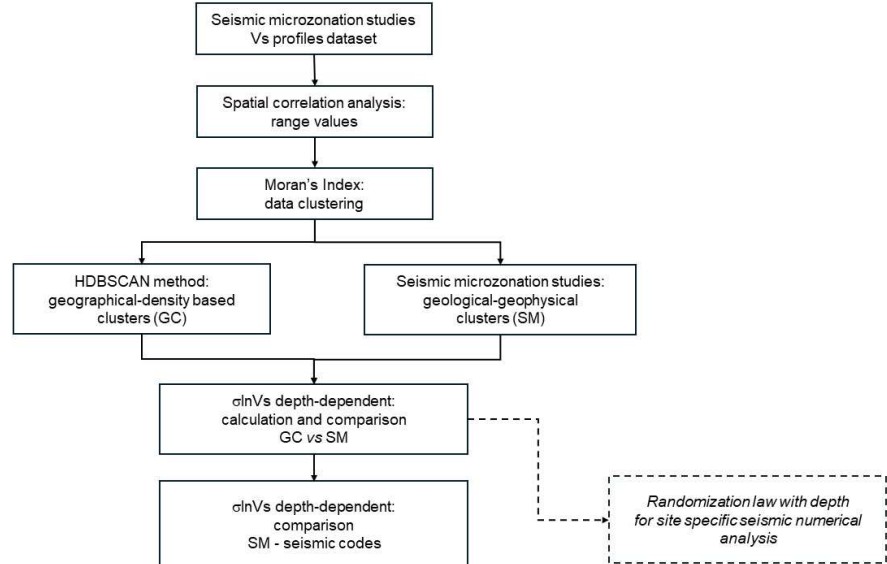

**Figure 1: Flow chart describing the steps described in the paper to define the σlnVs (standard deviation of Vs natural logarithm) depth-dependent values of the Vs profiles.**

## 2 Data and Methods

### 2.1 Vs profiles dataset

The original Vs database includes 23,512 Vs investigations classified as either punctual or linear. These investigations include Down-Hole (DH), Cross-Hole (CH), Extended Spatial Autocorrelation - Spatial Autocorrelation Phase Analysis (ESAC-SPAC), Multichannel Analysis of Surface Waves (MASW), Spectral Analysis of Surface Waves (SASW), Refraction





Microtremor (REMI), Seismic Refraction (SR), and Frequency-Time Analysis (FTAN). Table 1 shows a detailed analysis of the number of investigations by type and their corresponding geometries. All linear investigations are referenced to their center points to maintain consistency in data format.

| Geometry | Type of investigation | Count | % |
|---|---|---|---|
| Point | DH | 1,091 | 7.3 |
| Point | CH | 12 | 0.1 |
| Point | ESAC-SPAC | 753 | 5.1 |
| Line | MASW | 9,457 | 63.5 |
| Line | REMI | 3,160 | 21.2 |
| Line | SASW | 23 | 0.2 |
| Line | SR | 190 | 1.3 |
| Line | FTAN | 211 | 1.4 |

**Table 1: Breakdown of the number of Vs investigations for type and geometry.**


The dataset was refined by selecting investigations that reached a minimum depth of 30 meters and by removing outliers based on the interquartile range criterion applied to the natural logarithm of the Vs30 value. This filtering approach provided a refined dataset of 14,897 investigations (see supplementary material). The Vs profiles were discretized with a depth step size of 1 m. The obtained profiles include Vs values ranging from 75 to 2,034 m/s, with depths varying from 1 to 120 m (Fig. 2). The

hexbin plot, along with the marginal histograms, effectively displays the dataset by showing the frequency of Vs-depth pairs on a logarithmic scale, after removing outliers based on the interquartile range criterion applied to the natural logarithm of the Vs30 value (Fig. 2). The histograms provide a detailed overview of the distribution of Vs values and depths, independently.

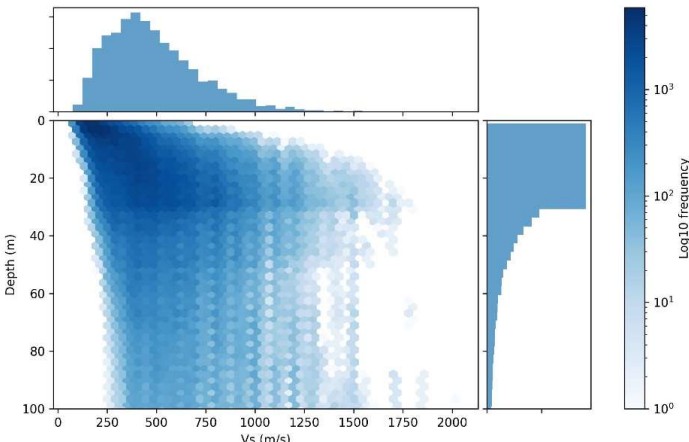

**Figure 2: Hexbin heatmap of Vs, and depth and frequency distribution of Vs values (top) and of depths (right).**





The spatial distribution of Vs30 values across Italy for the refined 14,897 Vs profiles is reported in Figure 3 left. The highest values of Vs30 are distributed in the mountain ranges while the lowest values are concentrated in the central-eastern region of the Po Valley.

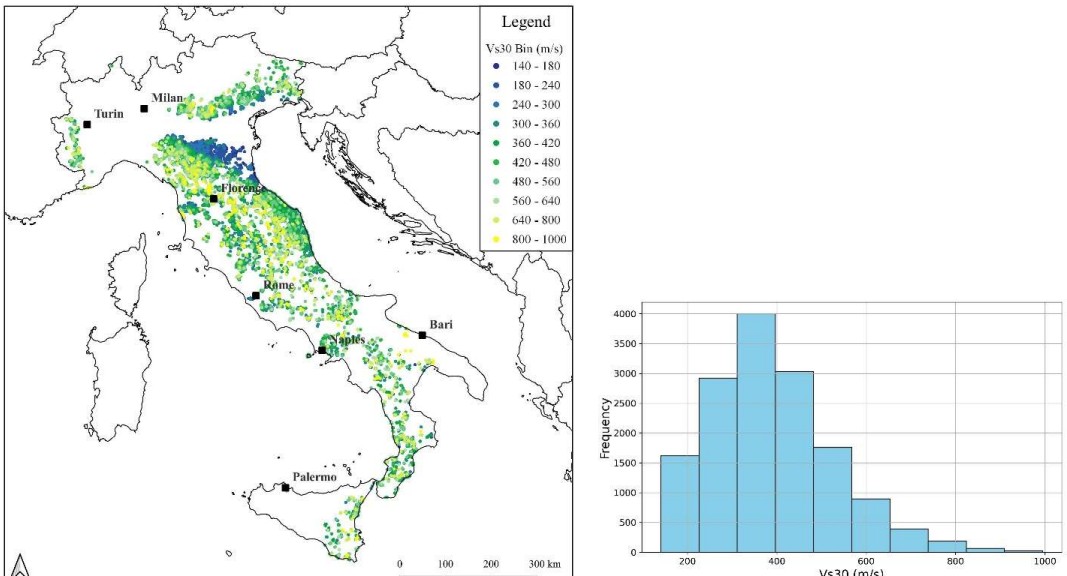

**Figure 3: Left - geographic distribution of the refined 14,897 Vs profiles across Italy in terms of Vs30; Right - frequency**
**distribution of Vs30.**

The latter information will be useful in discussing some of the results obtained. Furthermore, focusing on the frequency of these values, Figure 3 right shows that they are lognormally distributed.

**2.2 Spatial correlation analysis**

Experimental variograms were used to examine the spatial structure, such as autocorrelation, of the Vs data in terms of distance and variability.

The experimental variogram is a fundamental tool used in spatial statistics to quantify the spatial autocorrelation of regionalized variables, such as temperature, precipitation, or soil properties, across geographic space (Journel and Huijbregts, 1978; Deutsch

and Journel, 1998; Chiles and Delfiner, 2009). It provides insight into the relationship between the values of a variable based on distance or direction. The variogram quantifies the dissimilarity between pairs of observations at different locations based on their geographical separation. It describes the spatial relationship structure of the variable under study. The variogram function [1] is commonly referred to as γ(h), where h represents the lag distance or separation between two locations. The formula for the experimental (semi)variogram γ(h) can be expressed as Eq 1:




$$\gamma(h) = \frac{1}{2N(h)} \sum_{i=1}^{N(h)} [Z(x_i + h) - Z(x_i)]^2 \qquad (1)$$

where, $N(h)$ represents the number of pairs of observations separated by the lag distance $h$, $Z(x_i)$ denotes the value of the variable at location $x_i$, and $Z(x_i+h)$ is the value of the variable at a location $h$ units away from $x_i$. The sum is taken over all pairs of observations separated by the lag distance h.

The experimental semivariogram is typically computed by first dividing the study area into a set of lag intervals. For each lag interval, the average squared difference in values between pairs of observations separated by the corresponding lag distance is calculated. This process is repeated for multiple lag intervals, resulting in a curve representing the variogram function. For sample points at close distances, the difference in values between points tends to be small. In other words, the semi-variance is small. But when the sample points are further apart, they are less likely to be similar. This means that the semi-variance becomes large. As the distance from the sample points increases, there is no longer a relationship between the sample points. Their variance begins to flatten out, and the sample values are not related each other.

According to this spatial behaviour, the variogram curve typically exhibits three common descriptors and distinct parameters: i) the nugget ($C_0$), ii) the sill ($C-C_0$), and iii) the range (a) (Fig. 4).

The nugget ($C_0$): Theoretically, at zero separation distance (lag =0), the variogram value is 0. However, at an infinitesimally small separation distance, the variogram often exhibits a nugget effect, which is a value greater than 0. The nugget effect is a phenomenon present in many regionalized variables and represents short-scale randomness or noise in the regionalized variable typically caused by measurement error or micro-scale variability. It can be seen graphically in the variogram plot as a discontinuity at the origin of the function.

The sill ($C-C_0$): This is generally considered to be the variogram value at which the variogram curve flattens with increasing distance. The sill is also considered to be the variance of the data entering the variogram calculation. The sill (C) represents the plateau of spatial dependence and indicates the maximum achievable spatial correlation.

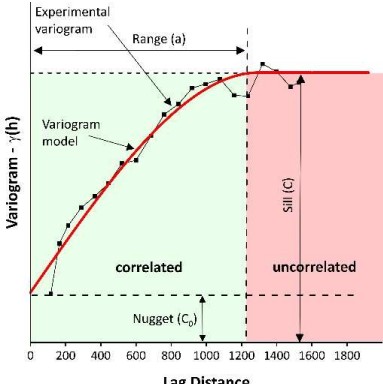

**Figure 4: Theoretical experimental semivariogram and model with corresponding parameters. For details, see text.**



The range (a) This parameter indicates the distance beyond which spatial correlation is negligible. Sample locations separated by distances closer than the range are spatially autocorrelated, whereas locations further apart than the range are not.

It can be evaluated valuable insight into the spatial structure and variability of regionalized variables by analysing the shape and parameters of the experimental variogram. The experimental variogram is usually fitted by a simple function (variogram model) using a mathematical formula whose main parameters (i.e. sill, range and nugget) are used to calculate values for

unsampled locations using the kriging technique (Goovaerts, 1997; Isaaks and Mohan Srivastava, 1989; Journel, 1987; Krige, 1966; Matheron, 1971;).

**2.3 Dataset Vs profiles clustering**

After the quantification of the spatial correlation between the Vs surveys, we checked whether the same surveys were clustered.

For this verification, we used the Moran's Index. Moran's Index quantifies spatial autocorrelation by considering both the locations and attributes of features simultaneously, and evaluates the distribution pattern (i.e., points or polygons) based on a set of features and their associated attribute by comparing the study pattern with standard clustered, dispersed, or random patterns. We used the Spatial Autocorrelation tool in ArcGIS Pro (ESRI.com). The programme calculates the Moran's I index value and provides a z-score and p-value to assess the importance of the index. P-values are numerical estimates of the area

under a given distribution curve, determined by the test statistic.

After checking whether the surveys in the dataset were clustered, we used two clustering methods. The first method involves a geographical analysis (i.e., location of the survey points) to define geographical clusters (GCs), while the second method is to consider SM polygons as clusters (SMs), extracted directly from the seismic microzonation projects.

*Geographic clustering: HDBSCAN method*

Geographical clustering considers the geographical location of the Vs surveys, and therefore the density of survey points in a given area, without reference to the Vs parameters. The clustering was performed using the algorithm HDBSCAN (Hierarchical Density-Based Spatial Clustering of Applications with Noise in McInnes et al., 2017) algorithm available in ArcGIS Pro 3.2.2 suite (Copyright© 2023, Esri. Inc.). HDBSCAN is a density-based clustering method that, thanks to the

concept of mutual reachability distance, is particularly able to manage issues related to the recognition of clusters with different densities. The process begins with the construction of a minimum spanning tree that incorporates this mutual reachability distance, followed by the development of a hierarchical tree of clusters. This methodological foundation allows the adaptive identification of clusters without the need for a predefined number of clusters, thus providing deeper insights into the intrinsic structures of data by combining density-based clustering with hierarchical analysis and stability metrics.


*Geological-geophysical clustering: seismic microzonation studies*

Another method of clustering Vs profiles involves the use of the boundaries of SM polygons (Figure 5). As mentioned in the introduction, the seismic microzonation project that is underway in Italy involves the identification of SMs, that are several



square kilometres in size, homogeneous in terms of expected seismic amplification, and defined by means of geological,
geotechnical, and geophysical information.

The 14,897 Vs profiles provided are associated with 7,583 SMs distributed throughout the Italian territory (updated to
December 2022). These SMs are characterized by a variable number of Vs profiles, ranging from 1 to several tens. In order to
assess the statistical significance of our classification, a spatial statistical analysis of the Vs profile within the SMs was
performed. For statistical purpose only, 1,271 SMs containing at least 3 investigations were considered, giving a total of 7,350
Vs profiles. Most of the selected SMs (about 900) are characterised by 3 to 5 Vs profiles, while a few SMs contain more than
25 Vs profiles with a maximum of 68 investigations.

A SM is typically characterized by an extent of up to 20 km$^2$ with and interquartile range of 6 km$^2$ and a median value of 4
km$^2$, while the Euclidean distance between the Vs profiles within individual SMs mainly ranges between few metres and 5.5
km, with a median value of 1.8 km and interquartile range of 1.5 km.




**Figure 5: Example of a geological profile showing the partition into three Seismic Microzones (SM), identified by means the geological information and the results of geotechnical and geophysical investigations. The drawing is not to scale.**

**3 Results**

**3.1 Definition of range value and clustering**

The experimental variograms of the three Vs synthetic measurements Vs10, Vs20, and Vs30 show a spatial structure fitted by
nested (spherical + exponential) semivariogram models (Fig. 6). The first spatial structure shows a range of about 4,500 m,
while the second structure shows a range of 25,000 m. Table 2 shows the semivariogram parameters of the three models for
the three analyzed Vs synthetic measurements.

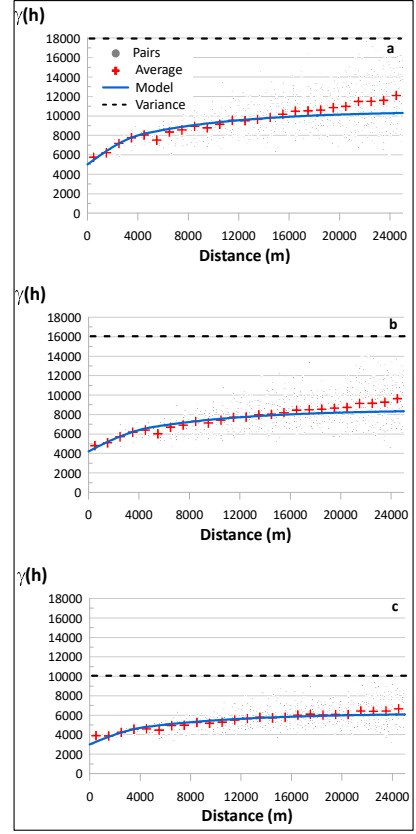

**Figure 6: Experimental semivariograms and nested models for a) Vs30, b) Vs20, and c) Vs10**

| Parameter | Nugget | Model 1 | Range 1 | Sill 1 | Model 2 | Range 2 | Sill 2 | Total Variance | Nugget/ Variance |
|---|---|---|---|---|---|---|---|---|---|
| Vs30 | 5000 | Sph | 4500 | 1500 | Exp | 25000 | 4000 | 18000 | 0.28 |
| Vs20 | 4200 | Sph | 5000 | 1000 | Exp | 25000 | 3300 | 16000 | 0.26 |
| Vs10 | 3000 | Sph | 4500 | 800 | Exp | 25000 | 2400 | 10000 | 0.32 |

**Table 2: Semivariogram parameters of the nested models (spherical + exponential) for the three Vs synthetic measures analyzed:**
**Vs30, Vs20, Vs10.**

It is useful to emphasize that our results are comparable with those results of Zhou et al. (2023), who carried out a quantitative study analyzing the lateral variation of the Vs profile and Vs30 in plain and piedmont terrains at short distances ranging from hundreds of metres to several kilometres. Even in their study, the variation in site conditions does not significantly increase

with distance within a specific range, usually between 1 km to 3-5 km.

The Morans' Index was calculated for the three synthetic parameters Vs30, Vs20, and Vs10, corresponding to the average of Vs in the first 30, 20, and 10 metres, respectively.



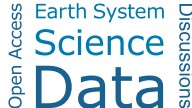

The results (Fig. 7) show a strong trend towards clustering of the data, so we applied the two clustering methods described in Method section.

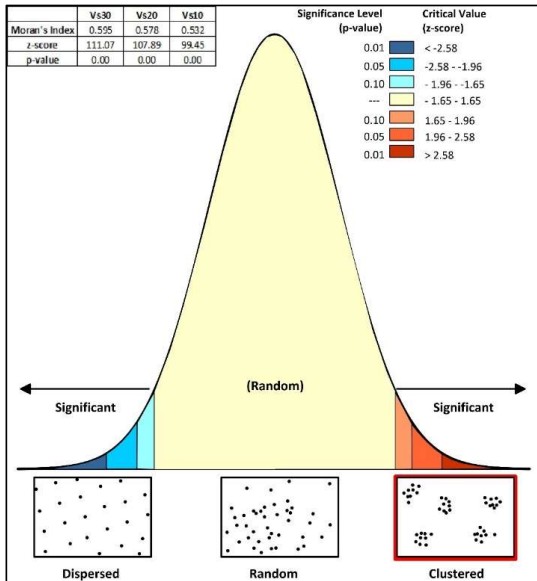


**Figure 7: Report of the Moran's Index (spatial autocorrelation) statistic showing that the pattern was clustered. p-Value: probability; z-score: standard deviation. for: a) Vs30, b) Vs20, c) Vs10**

The range value of 4,500 m of the first spatial structure was then used to filter clusters for both GC and SM, to obtain a high-

quality dataset for assessing the variability of Vs.

Table 3 shows the effect on the number of Vs profiles and clusters after applying two filters: minimum number of surveys in the cluster (i.e. 3) and spatial correlation distance (i.e., range value 4,500 m). The number of clusters obtained for the two methods (Tab. 3) is comparable (1759 vs. 1120 for GC and SM, respectively) and they are used to perform the final statistics on our target σlnVs.


| | Geographic clustering | | Seismic microzonation | |
|---|---|---|---|---|
| | *before filters* | *after filters* | *before filters* | *after filters* |
| Vs profiles (n.) | 12480 | 9601 | 14897 | 5561 |
| Clusters (n.) | 1977 | 1759 | 7583 | 1120 |

**Tab. 3: Number of Vs profiles and number of clusters obtained after applying the two filters: minimum number of surveys (i.e. 3) and spatial correlation distance (i.e., range value 4,500 m).**



Figure 8 shows the results of the distance distribution between Vs profiles after filtering with 3 minimum surveys and a range

value of 4,500 m: statistics are comparable (median values 500 m and 700 m for SM and GC clustering, respectively).

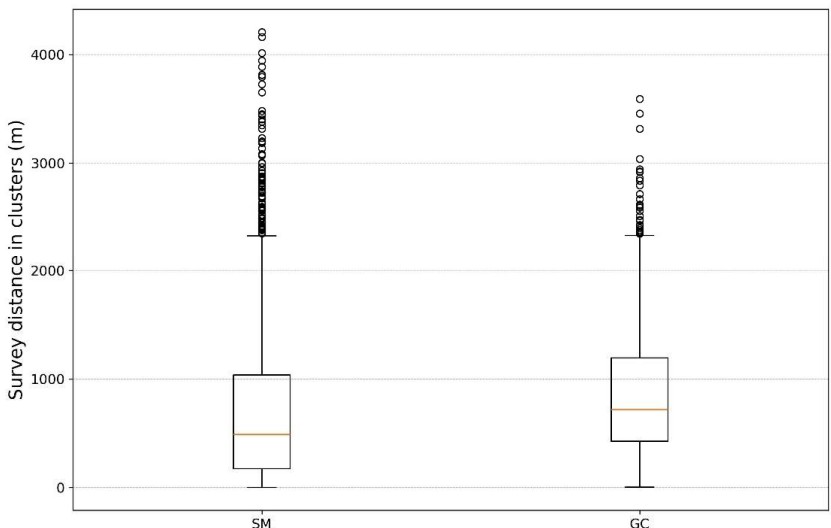

**Fig. 8: Distances distribution between Vs profiles after filtering with 3 minimum surveys and 4,500 m range value.**

### 3.2 σlnVs statistics in SM and GC clusters


After filtering the SM and GC clusters with the number of surveys and the range value of the first variogram structure, we

calculated the uncertainties as a function of depth. Uncertainty models assume that Vs values at each depth follow a lognormal

distribution (Toro, 2022 and references therein), therefore the uncertainties are determined by calculating the σlnVs.

Considering both SM and GC clustering, for the Vs profiles associated with the clusters:

240        • the σlnVs value was obtained for each meter of depth (Fig. 9 left)

• the percentiles $25^{th}$, $50^{th}$, $75^{th}$ of σlnVs were obtained for each meter of depth (Fig. 9 right).

The general trend of the curve in Figure 9 right, especially that of the 50th percentile, reflects a trend already known and often

reported in the literature (Toro, 2022 and references therein): shallower layers show greater variability and larger σlnVs values,

while Vs values remain almost constant below a depth of 50 m. The most interesting results are in the absolute σlnVs values:

the 50th percentile curve never exceeds the value of 0.21 and at depth below 50 m is about 0.11. Figure 9 right also shows

comparison between σlnVs values of SM clusters (red lines) and GC clusters (blue lines); it clearly shows the better





performance of SM for clustering the σlnVs values, which are always higher in the GCs, although the trend of the curve is very similar.

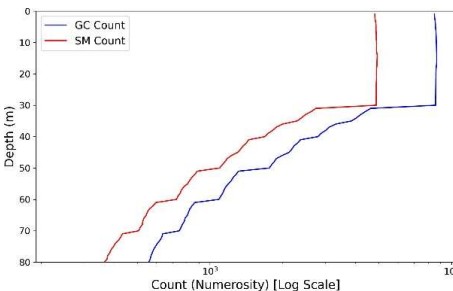 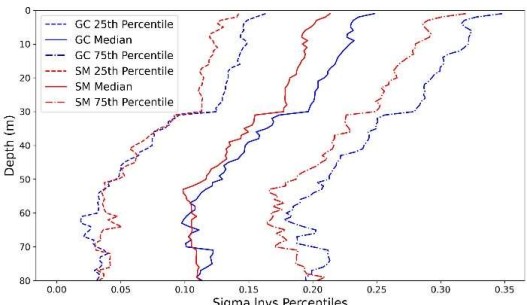

**Figure 9: Left - The numerosity of data per meter: very large for both methods, with a slight prevalence for GC method. The same figure shows that at about 30 m depth there is a variation in the number of data because many of the surveys do not investigate depths beyond 30 m; Right - σlnVs statistics: comparison among σlnVs values of SM and GC clustering (25th,50th,75th percentile).**

The difference between the uncertainty values in percent for the two methods is calculated by the following Eq. (2):


$$\text{Percentage Variation} = [(\,\sigma lnVs\_SM - \sigma lnVs\_GC)\,/\,\sigma lnVs\_SM]*100 \tag{2}$$

Where:

- σln(Vs$_{SM}$) is the standard deviation of the natural logarithm of the measurement obtained with the SM clustering.
- σln(Vs$_{GC}$) is the standard deviation of the natural logarithm of the measurement obtained with the GC clustering.

In general, there is a 14% decrease in favour of the SM method within the first 30 m, followed by a 9% reduction from 30 to 50 m, and a 4% decrease from 50 to 80 m. The decrease in the first 30 m, represents the quantification of the importance of geological and geophysical information in reducing uncertainties. Below 30 m, the difference in uncertainty between the two methods decreases to about 9% and then to 4%, due to the greater homogeneity of geological and geophysical properties at 265 greater depths.

**3.3 Comparison with seismic code and literature uncertainties**

We also compared our results with the uncertainties implemented in seismic codes and with known literature data (Figgs. 10 and 11). Appendix B of the EPRI-SPID (1993) provides guidance on the development of site response, including the 270 quantification of uncertainty. In cases where limited site response data are available, EPRI-SPID (1993) recommendations define aleatory uncertainty as lateral variations within a footprint of approximately 100 to 200 m. The EPRI-SPID (1993, Section B-4.1) recommends σlnVs values of 0.25 at the surface, decreasing to 0.15 at 15 m and deeper. Figure 10 shows the comparison of these values with the results obtained in the SM clusters also including the site-specific values from Toro (1995,

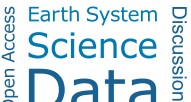

1997, 2022). The plot (Fig. 10) shows a clear improvement for the first 10-15 m, but in general all values of the depth

uncertainties from EPRI-SPID (1993) and Toro (1995,1997, 2022) are within the 25th and 75th percentiles of our results.

In earthquake-resistant standards (i.e., European EC8, NEHRP for the USA, and Italian NTC18), the dynamic characterisation of sites is represented by the synthetic value of Vs30. The Toro (1995) model is also developed for 4 NEHRP Vs-based soil classes (B: Vs30>760 to 1500, C: 360 to 760, D: 180 to 360, E: <180). Figure 11 shows σlnVs of Vs profiles from our high-quality dataset, reclassified according to the 4 NEHRP soil categories and considering a minimum number of 150 values for

each metre depth. Values of σlnVs constant with depth according to Toro (1995) are also shown in Figure 11 for reference. The following observations can be made.

- For soil category B (soil category A in the Italian building code), there is a strong variation with depth. This tendency is due to the strong heterogeneity of the rocks at the surface due to fracturing and weathering and, conversely, to the homogeneity of the rocks at greater depths.

- For soil categories C and D, the σlnVs values are fairly constant but generally larger to those obtained by SM clustering.

- For soil category E, the σlnVs values are low and comparable with those obtained by SM clustering (see Fig. 11). Considering that almost all the sites with these Vs30 values are located in the central eastern part of the Po Valley (Fig. 3), this is not surprising. The plot of uncertainties for soil category E somewhat describes a regional cluster, that

is much larger than an SM cluster but contains sites with similar Vs profiles.

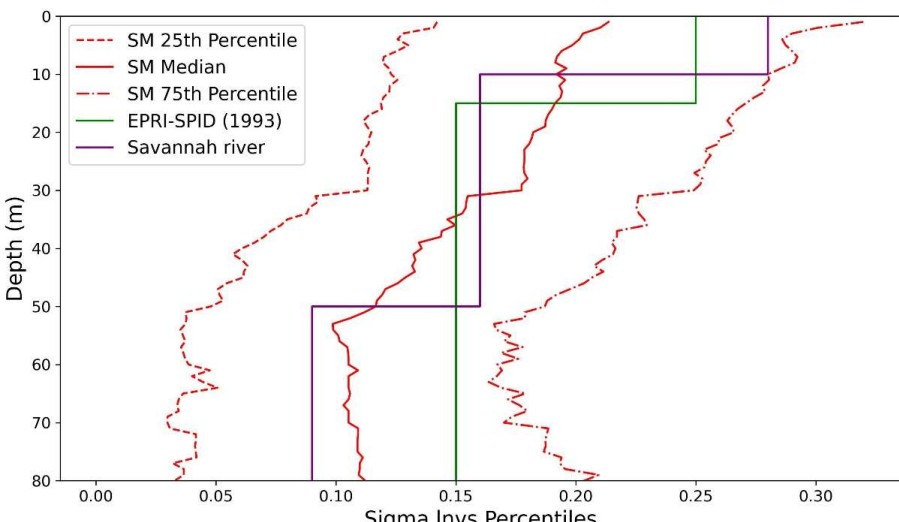

**Figure 10: Results σlnVs by means of SM clustering (red lines; 25,50,75th percentile) and comparison with EPRI-SPID (1993) (green line) and Savannah river nuclear project (Toro, 1995; 1997) (purple line).**



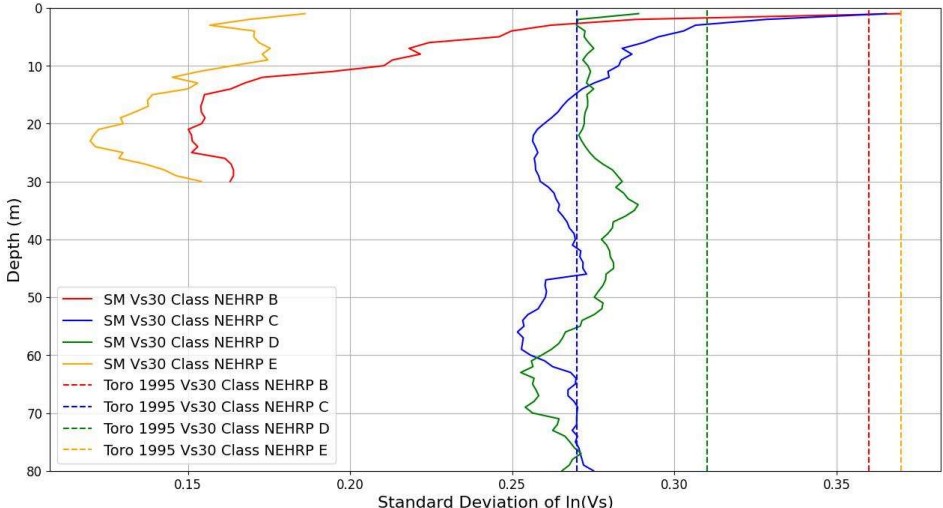


**Figure 11: The σlnVs values of Vs from our high-quality dataset, classified according to the 4 NEHRP soil categories; Toro (1995) constant σlnVs values are also reported.**


**4 Data availability**

The dataset is available at https://doi.org/10.5281/zenodo.10885590 (Mori et al., 2024). The records are:

- survey id;

- survey latitude and longitude in UTM33N coordinates;
- survey type;
- depth in meters;
- shear wave velocity value (Vs) in m/s;
- seismic microzonation (SM) cluster id.


**5 Conclusions**

We analysed the largest dataset of shear wave velocity profiles (Vs) currently available, comprising approximately 15,000 profiles, with the aim of providing insight into the uncertainty of the shear wave velocity Vs (σlnVs) for each metre depth and





up to a maximum of 80 m. A first spatial analysis showed that the synthetic Vs data (Vs10, Vs20, Vs30) show nested

semivariograms with a first spherical model and a second exponential model: the first spatial structure shows a range of about 4,500 m, while the second structure shows a range of 25,000 m. A second spatial correlation analysis using Moran's index revealed that the nature of the synthetic Vs data (Vs10, Vs20, Vs30) is inherently clustered, due to both the concentration of data in urban areas and the discrete nature of the geological bodies. This result led us to investigate the logarithmic standard deviation of the shear wave velocity (σlnVs) profiles in two different types of clusters: seismically homogeneous clusters based

on seismic microzones (SM), and non-seismically homogeneous clusters based only on geographic density (GC).

The variographic analysis allowed us to study σlnVs by filtering high quality clusters, specifically by imposing a maximum distance between surveys within clusters of 4,500 m, corresponding to the range value of the first spatial structure. The detailed study of σlnVs reveals the following significant statistical results:

- σlnVs in clusters that are homogeneous in terms of expected seismic behaviour (i.e., based on Seismic Microzones,
SM) are consistently lower values than those obtained from geographic density clusters (GC); the difference is 14% for the first 30 m, increasing to 2% at greater depths;
- σlnVs values in guidelines for site-specific analyses (EPRI - SPID, 1993) are internal to the percentiles of the SM statistics;
- σlnVs values in clusters are lower than those previously reported in the literature for soil Vs30 classification,
indicating the effectiveness of our methodological approach, while confirming the effectiveness of seismic microzonation as a tool for mapping in terms of expected site effects.

For practical applications, such as in numerical simulations to calculate seismic amplification, σlnVs values are essential to generate randomised velocity profiles. In this respect, our study supports the effective use of numerical simulation codes such as STRATA (Kottke et al., 2013) and the recently developed NC92soil (Acunzo et al., 2024).

In addition, our results have broader implications for:

- optimising borehole sampling designs in seismic microzonation projects;
- improving seismic hazard analysis, in terms of better management of uncertainties arising from the use of the parameter Vs.

Echoing Toro (2022), this study was inspired by the recommendation that "... *new site-specific stochastic Vs models should be*
*developed using these* (recent) *larger datasets, together with insights gained in research in the practical use of these models".* This highlights the value of using large datasets and recent research results to develop stochastic Vs models that can be used for site-specific applications and seismic hazard assessment.

Our contribution improves the discussion on seismic hazard calculation by addressing the complexities and uncertainties associated with Vs in ground motion models. It is crucial to thoroughly control Vs-related uncertainties in site response

analysis, as they significantly affect the understanding of earthquake ground motions. Our conclusions highlight the importance of paying close attention to uncertainties in seismic hazard assessment and contribute to the advancement the discipline in this area.



**Author contribution**

F.M. Conceptualization / Methodology/ Formal analysis/ Writing; G.N. Conceptualization /Methodology/ Formal analysis/ Writing; A.M. Data curation /Writing; G.C. Formal analysis/ Writing; C.V. Formal analysis / Writing; M.M. Conceptualization/ Supervision / Writing

**Competing interests**

The authors declare that they have no conflict of interest.

**Acknowledgements**

This work was supported by the Italian Department for Civil Protection of the Presidency of Council of Ministers within the "Accordo ai sensi dell'art. 15 legge 7 agosto 1990, n. 241 tra la Presidenza del Consiglio dei Ministri il Dipartimento della
Protezione Civile e il Consiglio Nazionale delle Ricerche Istituto di Geologia Ambientale e Geoingegneria per il supporto al Dipartimento della Protezione Civile per la realizzazione delle attività di cui all'ordinanza 780/2021 riguardanti gli interventi di prevenzione del rischio sismico, previsti dall'art. 11 del decreto-legge 28 aprile 2009 n. 39, convertito, con modificazioni, dalla legge 24 giugno 2009, n. 77, come rifinanziato dalla legge 30 dicembre 2018, n. 145 (CUP: B75F21002870001)" and within the "Accordo ai sensi dell'art. 15 legge 7 agosto 1990, n. 241 tra la Presidenza del Consiglio dei Ministri il Dipartimento
della Protezione Civile e il Consiglio Nazionale delle Ricerche Istituto di Geologia Ambientale e Geoingegneria per il supporto al Dipartimento della Protezione Civile per la programmazione degli interventi in materia di riduzione del rischio sismico ai fini di protezione civile (CUP: B53C22009330001)".

The activities of Amerigo Mendicelli were funded by "ICSC National Research Centre for High Performance Computing, Big Data and Quantum Computing (CN00000013, CUP B93C22000620006) within the European Union-NextGenerationEU
program" (Scientific manager for CNR - IGAG: Massimiliano Moscatelli).

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
