# Peer review of "Characterizing uncertainty in shear wave velocity profiles from the Italian seismic microzonation database"

_Earth System Science Data, 2024_

## Referee Comment (RC2)

This paper studies the variability of Vs profiles within seismic microzones and finds that the Vs scatter is larger when the microzones are defined on purely geographical properties (GEOGR) than when they are assessed by taking into account also geological and geophysical data (GEOL_GEOP). This should probably be an expected result, as the geophysical data, used to define the microzones in the GEOL_GEOPH case, should already include Vs profiles (am I correct?).

I think that the paper needs to be rethought and rewritten in some parts and that the calculations performed by the authors are not enough to support their final assessments.

**MAIN ISSUES**

As a general remark, I think that the use of the expression "Vs uncertainty" throughout the paper is misleading. By "Vs uncertainty" I understand the whole set of experimental/analysis/interpretation errors included in a Vs profile definition.
Here, it seems to me that by "Vs uncertainty" the authors mean only Vs variability at a specific depth among different Vs profiles provided by different authors. If this is the case, I suggest that the authors remove the word "uncertainty" from the entire paper and use only "variability" (starting from the title, that I find very misleading).

The abstract need to be entirely rewritten. It is hard to understand and very ambiguous (please see the comment in the attached text).

***Dataset.*** In line 58 I think the authors should properly address the real authors of that large amount of Vs profiles made available. This sentence "As a by-product, this work also provides for the first time the largest database of Vs profiles" suggests that the authors themselves compiled this entire Vs database while it seems to me that they are using data collected and processed by other people (the Mori et al. 2024 link does not say anything explicit about this).

Second, I am quite concerned about the presented dataset. On line 62 the authors state "we build a robust and reliable dataset of Vs profiles by removing any errors or duplicates (section 2.1);" but it seems to me that they did not check any data, analysis or interpretation quality. It seems to me that they just checked for double entries or similar. Is this the case? If so, this does not make the database neither "robust" nor "reliable".

I plotted the provided Vs profiles vs depth and got the picture below: a large number of Vs profiles reaches depths much larger than 30 m, even 120 m and it seems that the wide majority of these very deep Vs profiles were obtained by means of MASW or ReMi surveys. Honestly, in my 25 years experience I have never been able to achieve depths much larger than 20-30 m with MASW or ReMi and definitely not by using them alone (without combining them with H/V or other surveys). Just in a few lucky cases of soils without large impedance contrasts it is possible to go deeper than 20 m using conventional MASW/ReMi approaches.

I really need to be convinced that these Vs profiles reaching depths larger than 30-40 m are reliable. Please provide us with examples and convince us that such data are robust.

***Statistics.*** Parametric statistics based on means and standard deviations cannot be applied to datasets with only 3 values, which is the minimum threshold set by the authors. The authors claim that the Vs distribution at a specific depth for a specific zone is log-normal but they should show this at least for the cases where they have a large enough number of Vs values (i.e. more than 30 values). Fig. 3 shows the distribution of Vs30 values, which is something different from my present request.

The statistics applied by the authors does not make sense on database with less than 20-30 values. Please provide us with more information and maybe apply a non parametric statistics.

[Figure]

$\sigma(ln\ Vs)$ = 0.15 (am I properly understanding the parentheses?) means Vs deviations lower than $\pm15$ m/s (am I correct?). Is this real? Such precision cannot be reached even in the field at those depths (> 30 m), with experimental surveys. If the deviation of the mean of different Vs profiles for a given depth within a specific microzone is lower that 15 m/s (or something like that), than the experimental error in Vs assessment (which is certainly larger and is not considered in this paper) should also be taken into account.

The authors start discussing Vs profiles and Vs30. Then, at some point late in the paper, they introduce Vs10 and Vs20. Do we really need the latter two in this discussion?

With reference to line 268: An important point to address is understanding whether the Vs profiles collected for a same seismic microzone were produced by the same person (or few people) because this would mean that data are not fully independent and this generate a bias in their distribution. Please consider this point in the discussion. You are probably not dealing with truly independent datasets (at least from the interpretation point of view).

**MINOR ISSUES**

I found a little bit annoying the use of so many adjectives/nouns like "crucial", "significantly", "precision", "effectiveness", "discoveries", "unparalleled", "robust and reliable", "detailed", "valuable", particularly when it is not truly demonstrated in the text that such performances have been truly achieved. You can be convincing also without using all those adjectives.

[revised manuscript text omitted]

---

## Author Comment (AC1)

The Authors would like to thank the anonymous referees whose comments made it possible to improve the paper considerably.

In the document, the texts in light blue are the authors' answers.

Text in italics are parts of the modified or newly added text.

For further clarity on this note, the authors have completely rewritten abstracts and modified the workflow of Fig. 1.

**Ref 1**

The quality of the paper is generally fine. However, the 'results' section is quite underdeveloped, and the figures are not very good quality. The authors only report the statistics but do not provide their interpretation at length.

**General answer**

About this observation, we have actually mischaracterised the main statistical results of the paper. For example, we have replaced Figure 9 to make it clear that we analyzed the sigma lnvs trend in depth for all clusters considered after the filters (1120 clusters). So, the result is a statistic of the variation of sigma lnvs in depth in the form of percentiles.

To better clarify the novelty and importance of the results, the authors added a text to the conclusions entitled: "Interpretation of results and innovative contribution".

Considering the following points will greatly improve the technical soundness and readability of the paper:

1. Why are the Vs parameters being reported as 'Vs synthetic measurements' (e.g. Line 197). Are they simulated?

When the authors speak of "Vs synthetic", they mean calculated (Vs10, Vs20, Vs30) rather than measured.

We have replaced it with "*Vs calculated parameters*".

2. Why have Vs10, Vs20 and Vs30 been used for spatial correlation analysis? Why not average Vs at depth>30 m or H800?

This is because these are the same parameters analyzed in Zhou et al., 2023 in China and we wanted to compare and validate with another analysis. Adding the parameters H800 and Vs for depths >30m would not have led to statistically robust results such as those from using the data in the first 30m.

3. Why are the semivariogram statistics significantly different between the spherical and exponential models (Table 6 ? - see Figure 6 and Table 2)? Please provide a figure by showing both with respect to the data.

We have implemented the paper with:

*The semivariogram presented in Figure 6 (parameters in Table 2) is the result of the integration of two models: a first spherical model describing a scale between 1 km and 4.5 km and a second exponential model describing a larger scale between 4.5 km and 25 km.*

*The results we provide in terms of percentiles of sigma lnvs refer only to the clusters that respect the smaller scale (1-4.5km).*

*The concept of 'nested structures' in a semivariogram refers to the presence of multiple spatial processes that influence the variability of the data on different spatial scales. In other words, when analysing spatial data using a semivariogram, we may observe that spatial variability is not adequately described by a single semivariogram model but is better represented by a combination of models that capture variability at different scales.*

*These two models can be combined to form a nested semivariogram, where the overall structure of the semivariogram reflects the sum of the two components. The total semivariogram will have a more complex form than a single model but will provide a more accurate description of spatial variability on all scales considered.*

*How it is reflected in the semivariogram:*

*- At small scales (1-4.5 km): The variance initially increases rapidly, as described by the first model.*

*- At a larger scale (4.5-25 km): After the first range, the variance continues to increase but at a different rate, described by the second model.*

*The overall semivariogram will then be the sum of the two models and will show a transition between the dominance of the first model at a smaller scale and the second model at a larger scale. This combination better captures the complexity of the spatial processes affecting the data at different scales.*

4. Please improve the quality of the Figure 6. The symbol for the pairs is not visible at all. Please explain what is presented in this figure and how one can interpret the observations?

The theoretical part is explained in section 2.2. In any case, the grey dots represent all pairs of points at all distances determined by equation 1. The red crosses are the average value of gamma (eq.1) for all pairs of points in each class distance (i.e., the Lag).

The figure has been modified to make it clearer and more readable.

The text has been amended.

5. Please clarify how your results are comparable with Zhou et al. (2023) (Line 206). We know the range values you report are between 4.5 km and 25 km but how are they comparable with Zhou et al.?

We have replaced it with:

*Zhou et al. (2023) state that the variation in conditions between sites does not increase significantly with separation distance within a certain range (typically 1 km to 3-5 km). Table 2 shows that for the value of Vs30 (the same studied by Zhou et al.) the range (a parameter indicating the distance beyond which spatial correlation is negligible) is 25 km. We only consider and provide results on clusters with survey pair spacing of less than 4.5 km (reduced scale). In this sense, our results are comparable with the range of Zhou et al. (2023).*

6. Figure 7 is not legible at all. Please explain what is presented in the figure. The figure title mentions figure sublabels (a, b, c) but there are no such figures.

*The Moran's Index was calculated for the three parameters Vs30, Vs20, and Vs10, corresponding to the harmonic mean of Vs in the first 30, 20, and 10 meters, respectively. The results (Fig. 7) show that for all three parameters (Vs30, Vs20, Vs10), there is a strong positive spatial autocorrelation, which is indicated by the high Moran index and extreme z-score values. The z-score is an indication of the statistical significance of Moran's index. Very high values (as in this case) indicate strong evidence against the null hypothesis of no spatial autocorrelation. All p-values are 0.00, indicating that the observed spatial autocorrelation is highly significant and not due to chance. The central figure shows a normal distribution used to interpret the z-score. The central area (random) indicates no significant spatial autocorrelation, i.e. a random spatial pattern. Left and right sides (significant) indicate significant spatial autocorrelation. On the left-hand side, there is spatial dispersion (where points are more evenly distributed in space), while on the right-hand side (as in this case) there is concentration or clustering (where similar points tend to cluster together).*

Deleted a), b) c) from the caption.

7. As mentioned in Line 214, how does Figure 7 show a strong trend toward clustering? Please explain.

See response #6.

8. Please explain how to interpret observations from Figure 8 (median values 500 m and 700 m for SM and GC clustering)?

Line 231. Add: *The survey clustering criterion is different for the two methods: for the SM clusters, there is a geological, geomorphological and geophysical check to assess the size in depth. This control is not there in GC geographic clusters, which is why the median values of the distances between pairs of surveys are lower.*

9. Please provide a mathematical expression for σlnVs calculation.

About this observation, we have actually mischaracterised the main statistical results of the work. For example, we have replaced Figure 10 to make it clear that we analyzed the sigma lnvs trend in depth for all clusters considered after the filters (1120 clusters).

So, the result is a statistic of the variation of sigma lnvs in depth in the form of percentiles.

Replaced Fig.10 and caption with corrections.

The extended description of the expression σlnVs has been reported in the abstract (Line 16).

[Figure]

10. In Figure 9 (left), what does the X-axis signify? How can one interpret this figure? Please explain it in text rather than in the Figure title. Please make the X-axis ticks visible so that one can read the counts. Please scale both figures (left and right) to same size.

Figure 9 left in abscissa (log scale) presents the number of surveys per depth.

Replaced Fig.9 left and caption with corrections.

[Figure]

11. Please provide maps showing examples of SM and GC areas.

The authors give an example of an SM cluster and a GC cluster. This example is explanatory for Ref 1, which requested it, but is not given in the text of the paper.

[Figure]

In the example are shown from left to right and from top to bottom:

1. seismic microzonation of the Campotosto Lake large area (there are several villages)

2. geographical GC clusters of the same large area

3. seismic microzonation of the Campotosto village only, with the only seismic microzone (to which the SM cluster corresponds) that meets the filtering criteria highlighted

4. the same zone with the SM cluster of the surveys

5. the comparison between the two clustering modes (and the GC cluster surveys).

12. This reviewer questions the validity of Line 264. If this analysis has been performed for areas that include sites from soft basins to mountain topography, then 30 m depth cannot be representative of the entire dataset. This is also observed at the end in Figure 12 (9b?).

*Add: For SM clusters, the analysis was conducted by separating surveys on sites in 'soft' basins and surveys on hard sites. The non-differentiation occurs in GC clustering only as there is no geological and geophysical control. This observation is especially true for the first 30 meters as below this depth, most of the time, the rigid bedrock becomes common between soft and rigid sites (see the cartoon of Fig. 2).*

13. Please rewrite the interpretations in Line 260-265 as the sentences are not easy to follow. The line "The decrease in the first 30 m, represents the quantification of the importance" is misleading.

*Replaced sentence: σlnVs in clusters that are homogeneous in terms of expected seismic behaviour (i.e., based on Seismic Microzones, SM) are consistently lower values than those obtained from geographic density clusters (GC); the difference is 14% for the first 30 m, followed by a 9% from 30 to 50 m, and a 4% from 50 to 80 m;*

14. How are the authors sure that "The plot (Fig. 10) shows a clear improvement for the first 10-15 m"

(Line 274)? The lower values in uncertainty could be due to many reasons. One reason could be that probably the data used after filtering do not represent all geology types and hence, do not capture the variability. If the authors could present the map of filtered data, then we could verify it.

As can be seen from the attached figure, the distribution of used and unused surveys is very similar throughout the country, not only geographically but also in geological/geophysical contexts. This reassures us as to the full geological representativeness of the filtered cluster sample.

[Figure]

15. Please provide references for European EC8, NEHRP for the USA, and Italian NTC18.

EC8 and NTC18 texts were removed from the paper. NEHRP added.

16. Why is the value of Vs30 synthetic as mentioned in Line 277 ("the dynamic characterization of sites is represented by the synthetic value of Vs30") and in many other instances? Isn't it calculated from geophysical measurements?

When the authors speak of "Vs synthetic", they mean calculated (Vs10, Vs20, Vs30) rather than measured.

We have replaced it with "*Vs parameters*".

17. It's apparent from Figure 11 and the authors' explanation that the standard deviations obtained from this study by soil class is not generally comparable with those from Toro. This might be an indication of the site-specific nature or data-specific influence of Vs variation. Hence the conclusion in Line 329-331 is not soundly established. This question has not been resolved by this paper. The authors could probably discuss about it or indicate it as a limitation of the study.

We agree that the curves of Toro (1995) are not comparable with the curves obtained in this paper. For the reasons given by the referee and because the curves of Toro (1995) are elaborations of scattered surveys over the territory that respond only to a classification of Vs30. The curves in this paper are closely related to the geological-geophysical homogeneity of the territory.

We propose to remove the constant sigma curves of Toro (1995) from Figure 11.

[Figure]

Editorial comments:

Line 18: here defined "as"        corrected

Line 29-30: What does it mean "about 4,000 out of approximately 8,000"?    corrected: *around 4000 municipalities out of a total of 8000*

Line 153: The paragraph title "Dataset Vs profiles clustering" is confusing.    corrected: Vs profiles clustering

Line 187: "An" SM                corrected

Line 211: "Moran's"            corrected

Table 3: The fonts are mixed    corrected

Line 285: larger "than"          corrected

Line 325: It's confusing "the difference is 14% for the first 30 m, increasing to 2% at greater depths". Please rewrite it.        corrected:  *the difference is 14% for the first 30 m, followed by 9% from 30 to 50 m, and 4% from 50 to 80 m*

Line 327: What does it mean "σlnVs values ... are internal to the percentiles". Please rewrite it.
        Corrected: *The σlnVs values in the guidelines for site-specific analyses (EPRI - SPID, 1993) fall within the 25th-75th percentiles of the SM statistics.*

Line 346: contribute to the advancement "of" the discipline  corrected